# Importance of small vessel disease as a possible cause of sudden sensorineural hearing loss

**Chul Young Yoon**[1,2], **Junhun Lee**[1,2], **Tae Hoon Kong**[1,2,3], **Young Joon Seo**[1,3]*

**1** Research Institute of Hearing Enhancement, Yonsei University Wonju College of Medicine, Wonju, South Korea, **2** Department of Biostatistics, Yonsei University Wonju College of Medicine, Wonju, South Korea, **3** Department of Otorhinolaryngology, Yonsei University Wonju College of Medicine, Wonju, South Korea

* okas2000@yonsei.ac.kr

## Abstract

### Objective

Vascular disease like small-vessel disease (SVD) is the most likely cause among the potential causes of Sudden sensorineural hearing loss (SSNHL). Understanding the relationship between SVD and SSNHL is crucial for developing effective prevention and treatment strategies. To confirm the relationship between SVD and SSNHL, the effect of SVD is confirmed by focusing on the duration and recurrence of SSNHL.

### Methods

This article reports a retrospective observational study that investigated the relationship between SVD and SSNHL using the South Korea Health Insurance Review and Assessment Service (HIRA) database from 2010 to 2020. This retrospective observational study included 319,569 SSNHL patients between 2010 and 2020.

### Results

Participant demographics were controlled using Propensity Score Matching. The hazard ratios (HR) for the effect of SVD on the duration of SSNHL were 1.045 for the group with SVD before the onset of SSNHL and 1.234 for the group with SVD after the onset of SSNHL. SVD was statistically significant for the recurrence of SSNHL, with an odds ratio of 1.312 in the group with SVD compared to the group without SVD. The HR for the period until a recurrence in the group with SVD was 1.062.

### Conclusions

The study identified SVD as a possible cause of SSNHL and found that the duration of SSNHL increased only in the presence of SVD. SVD also affected the recurrence of SSNHL, with the recurrence rate being 1.312 times higher in the group with SVD.

**Data Availability Statement:** For privacy protection reasons and in accordance with domestic laws in Korea, registration of health insurance data is restricted. Therefore, we obtained authorization to use third-party data, specifically

health insurance data, to conduct our research. As a result, we request an exemption from the data registration process. Data cannot be shared publicly because of National health insurance data from Korea. Data are available from the National health insurance service Institutional Data Access / Ethics Committee (contact via https://nhiss.nhis.or.kr/bd/ay/bdaya001iv.do) for researchers who meet the criteria for access to confidential data.

**Funding:** The present study was grant-funded by three institutions supported by the Korean government. - "Regional Innovation Strategy (RIS)" through the National Research Foundation of Korea (NRF) funded by the Ministry of Education (MOE) (2022RIS-005) - the National Research Foundation of Korea (NRF) grant funded by the Korea government (MSIT) (No. NRF-2020R1A2C1009789) - The Korean Fund for Regenerative Medicine (21C0721L1) - The Commercialization Promotion Agency for R&D Outcomes (2023, 1711199152) The funders had no role in study design, data collection and analysis, decision to publish, or preparation of the manuscript.

**Competing interests:** The authors have declared that no competing interests exist.

## Introduction

Sudden sensorineural hearing loss (SSNHL) is defined as a difference of at least 30 dB in three consecutive frequencies within 3 days, and 90% of SSNHL cases are idiopathic [1]. The incidence of SSNHL is difficult to determine with certainty because of the low treatment persistence. The incidence of SSNHL is 11–77 per 100,000 per year [2]. SSNHL occurs regardless of sex and age and shows various severities [3]. Although the spontaneous recovery rate of SSNHL is as high as 32–65%, the cause and pathogenesis of SSNHL are still unknown [4]. Furthermore, there have been reports suggesting that the spontaneous recovery rate of SSNHL has recently been exaggerated [1]. SSNHL can cause permanent damage if treatment is delayed. Despite the need to recognize the disease and take immediate action, the etiology of SSNHL is ambiguous. Previous studies have suggested possible causes of SSNHL, including vascular, ear & endocrinal disorder, infection, autoimmune, cancer, trauma, and drugs adverse effect [1, 5–8].

Among the potential causes of SSNHL, vascular disease is the most likely. According to a 2012 review by Lin et al., acquired and hereditary cardiovascular factors are associated with an increased risk of developing SSNHL [7]. Small-vessel disease (SVD) refers to the narrowing or blockage of small blood vessels that supply vital organs such as the heart, brain, and ears.

Although the exact pathophysiology of SSNHL has not yet been elucidated, understanding the relationship between SVD and SSNHL is crucial for developing effective prevention and treatment strategies. Therefore, we defined vascular disease as SVD, based on previous studies. Vascular disease was investigated as a risk factor for SSNHL from 2010 to 2020 using SSNHL patient data extracted from the HIRA (Health Insurance Review and Assessment service) database. By exploring the potential mechanisms and risk factors underlying these conditions, we hope to shed light on the pathogenesis of SVD and improve clinical outcomes in patients.

## Materials and methods

### Data collection

South Korea's National Health Insurance (NHI) covers 98% of the population. Citizens who subscribe to the NHI are guaranteed expenditure coverage whenever they visit a hospital. Records of hospital visits are sent to the HIRA for review [9]. The HIRA data includes personal information, diagnosis information, and treatment information of the patients who visited the hospital [10]. The HIRA data used in this study were SSNHL patient data (H91.2, ICD-10; International Classification of Diseases-10), which were opened at the request of the researcher, and they include diagnosis records, medication information, and treatment information of SSNHL patients from 2010 to 2020. This study was approved by the Institutional Review Board of Yonsei University Wonju Severance Christian Hospital (Wonju, South Korea) (CR321316). We obtained data access permission from HIRA, and the period during which we could use the data was from 2022-08-04 to 2022-12-31. The approval numbers for this study obtained from HIRA are M20210430249 and M20221005002.

### Operational definition and study participant

SVD is a collection of the vascular diseases defined in this study. Each vascular disease was defined according to the rules defined in the study that analyzed the diseases (Table 1). Hypertension, diabetes mellitus, and dyslipidemia, which are the most common vascular diseases, as well as vascular diseases previously defined in South Korea's health insurance data, were included. Applicable vascular diseases include myocardial infarction, angina, transient ischemic attack, peripheral artery disease, thromboembolism, heart failure, stroke, ischemic stroke,

**Table 1. Operational definitions for small vessel disease and sudden sensorineural hearing loss selected in our study.**

| | Small vessel disease | | |
|---|---|---|---|
| **Disease** | **ICD-10** | **Drug / Procedure** | **Etc** |
| **Hypertension** [4] | I10, I11, I12, I13, I15 | Drug code | Admission ≥1 or outpatient department ≥2 OR At least once a year |
| **Diabetes mellitus** [4] | E10, E11, E13, E14 | Drug code | Admission ≥1 or outpatient department ≥2 OR At least once a year |
| **Dyslipidemia** [4] | E78 | Drug code | Admission ≥1 or outpatient department ≥2 OR At least once a year |
| **Myocardial infarction** [11] | I21.0, I21.1, I21.2, I21.3, I21.4, I21.9, I22.0, I22.1, I22.8, I22.9, I23.0, I23.1, I23.2, I23.3, I23.4, I23.5, I23.6, I23.8, I24.1, I25.2 | | |
| **Angina (both stable and unstable)** [12] | I20.0, I20.1, I20.8, I20.9 | | |
| **Transient ischemic attack** [12] | G45.0, G45.1, G45.2, G45.3, G45.8, G45.9, G46.0, G46.1, G46.2 | | Admission or outpatient department ≥1 |
| **Peripheral artery disease** [12] | I65.0, I65.1, I65.2, I65.3, I65.8, I65.9, I66.0, I66.1, I66.2, I66.3, I66.4, I66.8, I66.9, I70.0, I70.1, I70.2, I70.8, I70.9, I73.9, I74.0, I74.1, I74.2, I74.3, I74.4, I74.5, I74.8, I74.9 | Procedure code | Hospitalization ≥1 or outpatient department ≥2 |
| **Thromboembolism** [4] | I74 | | Hospitalization or outpatient department ≥1 |
| **Heart failure** [13] | I11.0, I50, I97.1 | | Hospitalization ≥1 or outpatient department ≥2 |
| **Stroke** [14] | I60-I63 | | |
| **Ischemic stroke** [14] | I63, I64 | Brain imaging (CT or MRI) ≥1 | Hospitalization ≥1 |
| **Cerebral infarction** [14] | I63 | Diagnostic imaging within 7 days (CT, MRA, MRI) | 1. ≥3 days of hospitalization or death within 3 days of hospitalization 2. Exclude other causes of brain injury |
| **Intracerebral hemorrhage** [13] | I60–I62, I64-6 | Brain imaging (CT or MRI) ≥1 | Hospitalization ≥1 |
| **Cerebral arterial stenosis** [14] | I65 or I66 | | |
| **Cerebral arterial dissection** [4] | I720, I725, I726, I670, or I671 | | |
| **Primary or secondary vasculitis** [4] | I68, L93, L95, M05, M30-32, M35, or D57 | | |
| **CADASIL** [4] | I673 | | |
| **Sickle cell disorder** [4] | D57 | | |
| **Retina vein occlusion** [15] | H34.1, H34.8 | | |
| **Diabetic retinopathy** [16] | E11.319 | Fundus photography | |
| **Chronic kidney disease** [4] | N18, N19 | | Hospitalization ≥1 or outpatient department ≥2 |
| | Sudden sensorineural hearing loss | | |
| **Sudden sensorineural hearing loss** [10] | H91.2 | Steroid drugs code ≥1 Pure tone audiometry test code ≥2 | |

CADASIL: cerebral arterial dissection, vasculitis, cerebral autosomal dominant arteriopathy with subcortical infarcts and leukoencephalopathy ICD-10: International classification of diseases-10, CT: Computed tomography, MRI: Magnetic resonance imaging, MRA: Magnetic resonance angiography

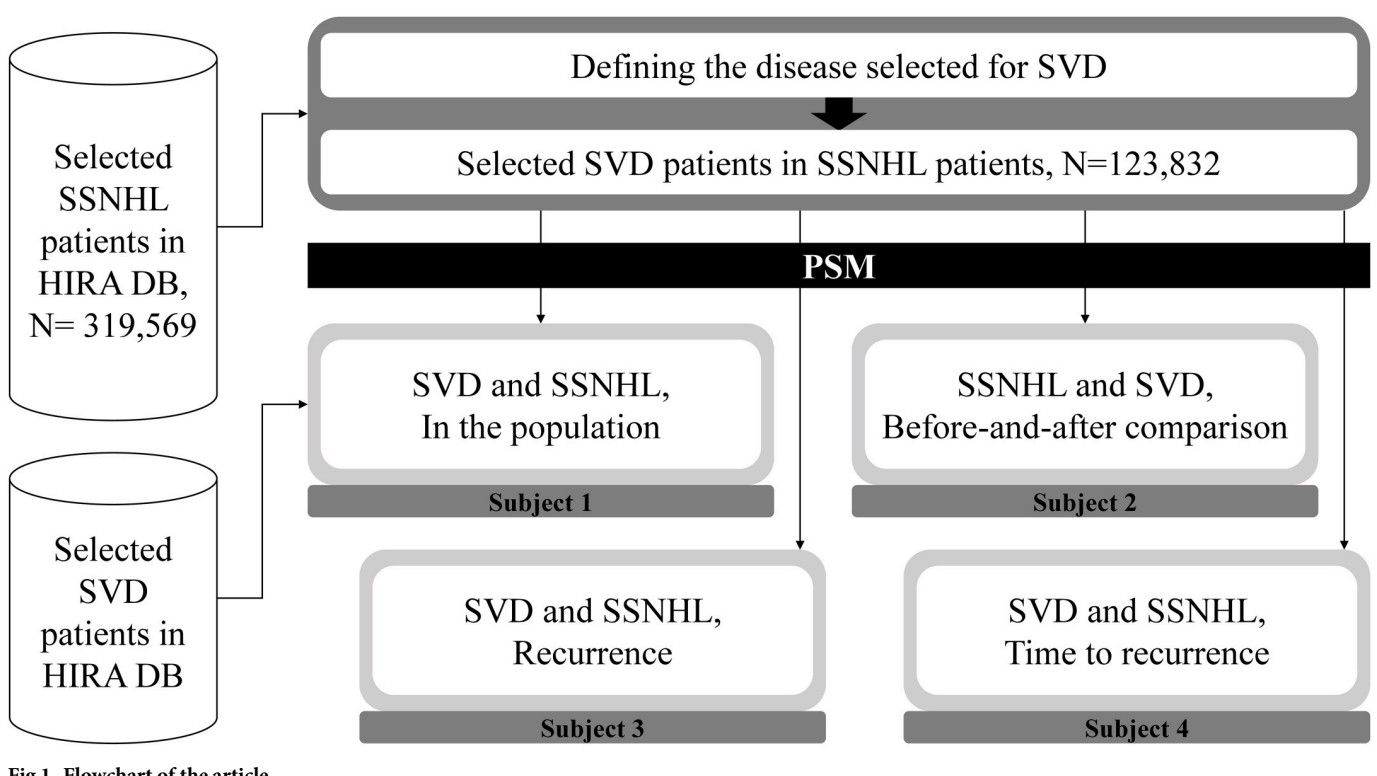

**Fig 1. Flowchart of the article.**

cerebral infarction, intracerebral hemorrhage (ICH), cerebral arterial stenosis (CASTN), cerebral arterial dissection, vasculitis, cerebral autosomal dominant arteriopathy with subcortical infarcts and leukoencephalopathy (CADASIL), sickle cell disorder, retinal vein occlusion, diabetic retinopathy, and chronic kidney disease. SSNHL was defined based on previous studies and the conditions were a H91.2 diagnosis code (ICD-10), one or more steroid drug codes, and two or more pure tone audiometry test codes. The Drug/Procedure codes in Table 1 can be found in the S1–S5 Tables.

When these conditions were applied, the number of patients diagnosed with SSNHL between 2010 and 2020 was 319,569. Diagnosis of SVD was confirmed in 126,836 patients.

## Experimental design and statistics

This study aimed to confirm the role of SVD as a risk factor of SSNHL. Therefore, we conducted the study by setting subjects based on various risk factors (Fig 1).

First, we extracted SSNHL patients from the HIRA database and defined SVD. The extracted patients numbered 319,569, with 123,832 having a history of SVD. Since both SSNHL and SVD, which are the focus of our study, are conditions heavily influenced by age, it was necessary to control for demographic characteristics. Therefore, we controlled for gender and age through Propensity Score Matching (PSM, Table 2). However, as the target groups varied depending on the subject, PSM was conducted for each subject. In Subject 1, we compare the ratio of SVD between the entire population of South Korea and the SSNHL patient group from 2010 to 2020. In Subject 2, we examine the impact of the duration of the disease on SVD among SSNHL patients. Subject 3 investigates the influence of SVD on SSNHL recurrence, while Subject 4 explores the impact of SVD on the time until SSNHL recurrence.

**Table 2. Result of PSM for each subject's data.**

(A) The results of conducting PSM on Subject 1

| Subject 1 | | Before matching | | | After matching | | |
|---|---|---|---|---|---|---|---|
| | | In SSNHL | Nationwide | p | In SSNHL | Nationwide | p |
| Sex | Male | 0.449 | 0.498 | < .0001 | 0.498 | 0.498 | 0.9684 |
| | Female | 0.551 | 0.502 | | 0.502 | 0.502 | |
| Age | 0~19 | 0.050 | 0.172 | < .0001 | 0.171 | 0.172 | 0.9792 |
| | 20~29 | 0.112 | 0.132 | | 0.133 | 0.132 | |
| | 30~39 | 0.171 | 0.132 | | 0.132 | 0.132 | |
| | 40~49 | 0.240 | 0.159 | | 0.158 | 0.159 | |
| | 50~59 | 0.279 | 0.166 | | 0.166 | 0.166 | |
| | 60~ | 0.148 | 0.239 | | 0.239 | 0.239 | |

(B) The results of conducting PSM on Subject 2

| Subject 2 | | Before matching | | | | After matching | | | |
|---|---|---|---|---|---|---|---|---|---|
| | | After SSNHL | Before SSNHL | Not SVD | p | After SSNHL | Before SSNHL | Not SVD | p |
| Sex | Male | 0.482 | 0.486 | 0.425 | < .0001 | 0.425 | 0.425 | 0.424 | 0.968 |
| | Female | 0.518 | 0.514 | 0.575 | | 0.575 | 0.575 | 0.576 | |
| Age | 0~19 | 0.014 | 0.003 | 0.079 | < .0001 | 0.079 | 0.080 | 0.079 | 0.979 |
| | 20~29 | 0.044 | 0.013 | 1.725 | | 0.173 | 0.172 | 0.173 | |
| | 30~39 | 0.142 | 0.054 | 0.233 | | 0.233 | 0.233 | 0.233 | |
| | 40~49 | 0.305 | 0.177 | 0.260 | | 0.260 | 0.260 | 0.260 | |
| | 50~59 | 0.362 | 0.432 | 0.190 | | 0.190 | 0.190 | 0.190 | |
| | 60~ | 0.134 | 0.321 | 0.065 | | 0.065 | 0.065 | 0.065 | |

(C) The results of conducting PSM on Subject 3

| Subject 3 | | Before matching | | | After matching | | |
|---|---|---|---|---|---|---|---|
| | | With SVD | With not SVD | p | With SVD | With not SVD | p |
| Sex | Male | 0.453 | 0.413 | < .0001 | 0.413 | 0.413 | 1.000 |
| | Female | 0.547 | 0.587 | | 0.587 | 0.587 | |
| Age | 0~19 | 0.050 | 0.054 | < .0001 | 0.054 | 0.054 | 1.000 |
| | 20~29 | 0.114 | 0.102 | | 0.102 | 0.102 | |
| | 30~39 | 0.170 | 0.172 | | 0.172 | 0.172 | |
| | 40~49 | 0.239 | 0.245 | | 0.245 | 0.245 | |
| | 50~59 | 0.278 | 0.288 | | 0.288 | 0.288 | |
| | 60~ | 0.149 | 0.139 | | 0.139 | 0.139 | |

(D) The results of conducting PSM on Subject 4

| Subject 4 | | Before matching | | | After matching | | |
|---|---|---|---|---|---|---|---|
| | | With SVD | With not SVD | p | With SVD | With not SVD | p |
| Sex | Male | 0.381 | 0.450 | < .0001 | 0.450 | 0.450 | 1.000 |
| | Female | 0.619 | 0.550 | | 0.550 | 0.550 | |
| Age | 0~19 | 0.093 | 0.007 | < .0001 | 0.007 | 0.007 | 0.993 |
| | 20~29 | 0.198 | 0.023 | | 0.023 | 0.023 | |
| | 30~39 | 0.247 | 0.082 | | 0.082 | 0.082 | |
| | 40~49 | 0.265 | 0.221 | | 0.221 | 0.221 | |
| | 50~59 | 0.174 | 0.425 | | 0.424 | 0.425 | |
| | 60~ | 0.054 | 0.242 | | 0.242 | 0.242 | |

SSNHL: Sudden sensorineural hearing loss

SVD: Small-vessel disease, SSNHL: Sudden sensorineural hearing loss

SVD: Small-vessel disease

Regarding Subject 1, PSM was conducted with the SSNHL patient group and the population in South Korea (Table 2A). In Subject 1, we compared the ratio of SVD between the overall population and the SSNHL patient group through chi-square analysis. However, due to insufficient computing power to extract all patients with SVD from the entire HIRA dataset, we instead examined the ratios for each vessel disease included within SVD. Regarding Subject 2, patients were divided into a group with SVD before onset of SSNHL, a group with SVD after onset of SSNHL, and a group without SVD, and PSM was performed (Table 2B). The duration of SSNHL was defined as the difference between the last diagnosis date and the first diagnosis date. To prevent inclusion of recurrent cases, only cases with a difference of 365 days or less between the diagnosis dates were considered. The first diagnosis date of SSNHL was set as the event point for survival analysis. For Subject 3, PSM was performed by categorizing the entire population into groups with and without SVD (Table 2C). Recurrence in SSNHL was defined when the difference between diagnosis dates was 365 days or more. Logistic regression was conducted to examine the presence of SVD between groups with and without recurrence. In Subject 4, the recurrence group was further divided into groups with and without SVD, and PSM was conducted (Table 2D). The time until recurrence was defined as the difference between the first recurrence diagnosis date and the previous diagnosis date. The recurrence date of SSNHL was set as the event point for survival analysis. PSM was applied to all groups corresponding to each participant to control for demographic characteristics (sex and age), with the effects of sex and age removed from all groups. The statistical program used for the analysis was SAS version 9.4 (SAS Institute Inc., Cary, NC, USA).

## Results

### Compare the ratio of SVD between the entire population and the SSNHL patient (Subject 1)

Table 3 shows the results of the confidence interval estimation of SVD in the total population of South Korea as a proportion of SVD in the SSNHL group. It is possible to verify each SVD individually in the NHI database, but it is not possible to identify duplicate patients. Therefore, the expression is presented as a ratio for each SVD condition. Based on the estimated results, two conditions were selected as the exclusion criteria. The conditions were: (1) statistical insignificance between the SVD ratio in the national record and SSNHL and (2) an insufficient the number of participants. Diseases excluded using these conditions included ICH, CADASIL, and sickle cell disorder. These diseases were not analyzed further. The proportion of ICH did not differ between the SSNHL group and the total population. However, there were no sufficient cases of CADASIL and sickle cell disorder for the study.

### The influence of SVD on the SSNHL group (Subjects 2, 3, and 4)

The effect of SVD on the duration of SSNHL (Subject 2) was also investigated. Using the group without SVD as a reference, the hazard ratio (HR) of the group with SVD before the onset of SSNHL was 1.045, and the HR of the group with SVD after the onset of SSNHL was 1.234. The differences among the three groups were significant, but the HR for the group with previous SVD in the SSNHL group was not statistically significant (Table 4, Fig 2A).

We examined the effect of SVD on the recurrence of SSNHL (Subjects 3 and 4). In the logistic regression analysis, which confirmed the effect of SVD on the recurrence of SSNHL in the entire population, SVD was statistically significant. The odds ratio for recurrence was 1.312 in the group with SVD compared with the group without SVD (Table 5A). Survival analysis to examine the effect of SVD on the period until recurrence in the group with SSNHL recurrence

**Table 3. Comparison of SVD disease in patients with SSNHL and in the population of South Korea.**

| Small Vessel Disease | SVD ratio in National record | SVD ratio and confidence interval in SSNHL |
|---|---|---|
| Hypertension* | 0.133 | 0.151 (0.1483, 0.1532) |
| Diabetes mellitus* | 0.069 | 0.029 (0.0279, 0.0301) |
| Dyslipidemia* | 0.044 | 0.204 (0.2008, 0.2063) |
| Myocardial infarction* | 0.003 | 0.01 (0.0094, 0.0108) |
| Angina (both stable and unstable) * | 0.014 | 0.088 (0.0858, 0.0896) |
| Transient ischemic attack* | 0.003 | 0.032 (0.0308, 0.0332) |
| Peripheral artery disease* | 0.009 | 0.003 (0.0023, 0.003) |
| Thromboembolism* | 0.000 | 0.005 (0.0048, 0.0058) |
| Heart failure* | 0.005 | 0.036 (0.0344, 0.0369) |
| Stroke* | 0.012 | 0.038 (0.0364, 0.039) |
| Ischemic stroke* | 0.010 | 0.016 (0.0153, 0.017) |
| Cerebral infarction* | 0.010 | 0.009 (0.0082, 0.0094) |
| Intracerebral hemorrhage | 0.002 | 0.003 (0.00204, 0.0031) |
| Retina vein occlusion* | 0.001 | 0.007 (0.006, 0.0071) |
| Diabetic retinopathy* | 0.001 | 0.007 (0.006, 0.007) |
| Cerebral arterial stenosis * | 0.003 | 0.022 (0.0212, 0.0232) |
| Cerebral arterial dissection* | 0.002 | 0.012 (0.0113, 0.0128) |
| Primary or secondary vasculitis* | 0.005 | 0.02 (0.0186, 0.0205) |
| CADASIL** | 0.000 | 0 (0.0001, 0.0003) |
| Sickle cell disorder** | 0.000 | 0 (0.0001, 0.0003) |
| Chronic kidney disease* | 0.000 | 0.018 (0.0175, 0.0193) |

SVD: Small-vessel disease, SSNHL: Sudden sensorineural hearing loss, CADASIL: cerebral arterial dissection,

vasculitis, cerebral autosomal dominant arteriopathy with subcortical infarcts and leukoencephalopathy

* Statistically significant (p < 0.0001)

** The number of participants is insufficient

**Table 4. Results of analysis of the effect of SVD on the treatment period of SSNHL.**

| The treatment period of SSNHL | N | % | Censored (%) | Log Rank | HR | 95% CI | | p-value |
|---|---|---|---|---|---|---|---|---|
| With Not SVD | 5,788 | 41.80 | 0.09 | < .0001 | Ref | - | - | - |
| With before SVD | 2,754 | 19.89 | 0.36 | | 1.045 | 0.999 | 1.094 | 0.056 |
| With after SVD | 5,305 | 38.31 | 0.18 | | 1.234 | 1.188 | 1.281 | < .0001 |

SVD: Small-vessel disease, SSNHL: Sudden sensorineural hearing loss, HR: Hazard ratio, CI: Confidence Interval

and Cox regression showed that the HR of the group with SVD was 1.062, with the group without SVD used as the reference (Table 5B, Fig 2B).

## Discussion

The exact pathophysiology of SVD in SSNHL is not fully understood; however, several mechanisms have been proposed. The anterior inferior cerebellar artery (AICA) branches from the basilar artery and supplies blood from the anterior to the middle cerebellum. It is the sole vascular supply for the labyrinth, cochlea, and vestibular organs. Given that the labyrinthine

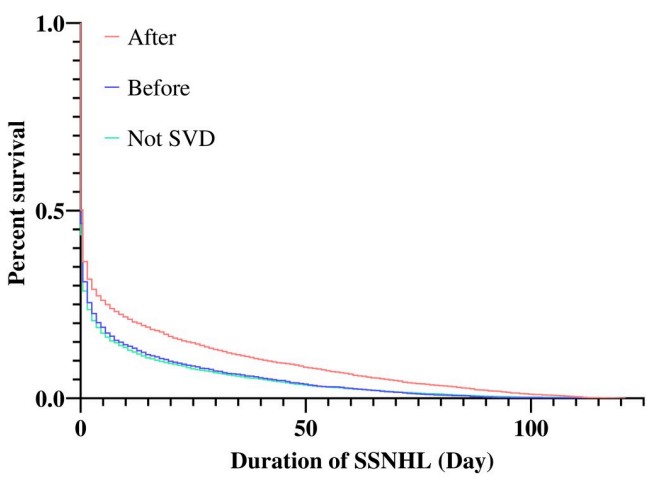

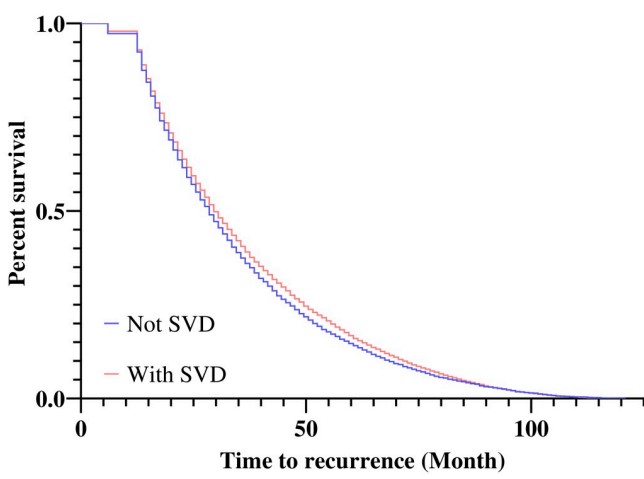

**Fig 2. Kaplan-Meier estimation of SSNHL duration and recurrence duration according to SVD.**

**Table 5. Result of analysis of the effect of SVD on the recurrence of SSNHL.**

(A) Result of subject 3

| Recurrence | N | % | p-value | Variable | OR | 95% CI | | p-value |
|---|---|---|---|---|---|---|---|---|
| No | 104,826 | 75 | < .0001 | Not SVD | Ref | - | - | - |
| Yes | 34,942 | 100 | - | With SVD | 1.312 | 1.281 | 1.345 | < .0001 |

(B) Result of subject 4

| Time to recurrence | N | % | Censored (%) | Time (month) | | Log Rank | HR | 95% CI | | p-value |
|---|---|---|---|---|---|---|---|---|---|---|
| | | | | Mean | Median | | | | | |
| Not SVD | 4,093 | 41.80 | 0.42 | 35.84 | 29 | 0.0007 | Ref | - | - | - |
| With SVD | 15,848 | 19.89 | 0.44 | 37.41 | 30 | | 1.062 | 1.026 | 1.099 | 0.0006 |

OR: Odds ratio, CI: Confidence Interval

SVD: Small-vessel disease, HR: Hazard ratio, CI: Confidence Interval

artery is the end artery of the AICA, it is plausible that differences in the anatomical variations of the AICA result in the clinical findings of SSNHL. One possible mechanism is that SVD in the inner ear can lead to decreased blood flow and ischemia, which can damage the hair cells and neural structures responsible for hearing [17]. Another proposed mechanism is that SVD can cause inflammation and oxidative stress in the inner ear, leading to further damage to the hearing structures [18]. Studies have also suggested that SVD in the inner ear may be a manifestation of a more generalized vascular pathology that affects multiple organs, including the brain and heart [19]. This may explain the observed association between SSNHL and other cardiovascular diseases such as myocardial infarction and stroke. Overall, the pathophysiology of SVD in SSNHL is likely multifactorial, involving complex interactions between the vascular, inflammatory, and oxidative stress pathways. SVD has been linked to several other diseases that share characteristics with SSNHL, including stroke and dementia. SVD in the brain can cause lacunar infarcts, which are small, deep brain strokes that result from the occlusion of small blood vessels [20]. Strokes can lead to cognitive impairments, including memory loss, executive dysfunction, and language deficits, which are also observed in SSNHL [21]. In

addition, studies have found that patients with SSNHL are at an increased risk of stroke and transient ischemic attack [22], suggesting a shared underlying vascular pathology. Similarly, SVD in the brain has been linked to the development of dementia, particularly vascular dementia, which is caused by a series of small strokes and microinfarcts [23]. The cognitive symptoms of vascular dementia, such as impaired memory and attention, are similar to those seen in SSNHL [24]. Furthermore, studies have found that patients with SSNHL are at increased risk of developing dementia, particularly vascular dementia [25], further suggesting a shared vascular mechanism.

The HIRA database offers several advantages. The HIRA data were obtained from South Korea's NHI database. Since South Korea's NHI covers 98% of the population, it has a great quantitative advantage over data from a single center [10]. However, it is difficult to access all data for the entire period when the NHI existed in South Korea, owing to limitations in hardware, time, and money for analysis. Nevertheless, we assessed data on the SSNHL group during the 2010–2020 period. Because the HIRA database contains information on tests, procedures, and medications, including diagnostic codes, there is sufficient information to define the disease. Therefore, it is desirable to design a study using the appropriate criteria. We have defined the corresponding diseases in Table 1 and using supplementary material. While efforts were made to define the duration of SSNHL and recurrence as objectively as possible, bias inevitably arises due to the researcher's influence. Recurrence was defined as cases where the time between diagnoses was within 365 days; however, in some instances, patients may seek hospital care for the management of SSNHL beyond the 365 days after the initial diagnosis, indicating a conceptual challenge in defining recurrence.

Table 3 presents a comparison of the SVD ratios between the entire South Korean population and patients with SSNHL in Subject 1. The ratios and confidence intervals demonstrate notable differences in the prevalence of various vascular conditions between the general population and SSNHL patients. The analysis in Table 4 investigates the effect of SVD on the duration of SSNHL in Subject 2. The HR suggest that, compared to the group without SVD, the group with SVD before and after SSNHL onset exhibits significant differences. However, the HR for the group with previous SVD in the SSNHL group does not reach statistical significance. In Table 5, the logistic regression analysis for Subject 3 shows a statistically significant odds ratio for SSNHL recurrence in the group with SVD compared to the group without SVD. For Subject 4, the Cox regression analysis demonstrates a significant hazard ratio for time to recurrence in the group with SVD compared to the reference group. The results underscore the importance of understanding the association between SVD and SSNHL, suggesting potential avenues for prevention and intervention. The observed variations in SVD prevalence between the general population and SSNHL patients could prompt further investigation into the role of vascular factors in SSNHL development. In our study, SVD (excluding ICH, CADASIL, and sickle cell disorder) clearly differed in proportion in the SSNHL-affected population. SVD affects the duration of SSNHL, whether before or after its onset, and this effect indicates an increase in disease duration. In particular, the risk of developing SVD after SSNHL onset reached 1.234, which was statistically significant. The effect of SVD on SSNHL recurrence was statistically significant, and recurrence occurred 1.312 times more frequently in the group with SVD than in the group without SVD. An investigation of the period until SSNHL recurrence showed that there was a statistically significant difference in the period until recurrence in the presence or absence of SVD in the group with recurrence, with a risk of 1.062.

The clinical implications of the association between SVD and SSNHL are significant, as they suggest that prevention, early detection, and treatment of vascular risk factors may help reduce the risk of SSNHL. One potential avenue for prevention is management of

hypertension, which is a major risk factor for SVD [20]. Several studies have found an association between hypertension and SSNHL, and it has been suggested that controlling blood pressure may reduce the risk of SSNHL [22, 26] (3). Lifestyle modifications such as exercise and a healthy diet may also help reduce the risk of hypertension and SVD. Early detection of SVD may be possible through imaging studies, such as magnetic resonance imaging (MRI) or computed tomography (CT). These techniques can reveal signs of SVD in the brain, which may be early indicators of vascular pathology in other organs, including the inner ear [27]. Early detection could allow the implementation of interventions to slow the progression of SVD and reduce the risk of SSNHL. The treatment of SVD may involve a combination of lifestyle modifications and medication. Blood pressure-lowering medication, such as angiotensin-converting enzyme inhibitors and calcium channel blockers, may help prevent or slow the progression of SVD. Antiplatelet therapies, such as aspirin, may also be used to reduce the risk of thrombotic events associated with SVD. In addition, lifestyle modifications such as exercise, healthy diet, and smoking cessation may help reduce the risk of SVD and associated conditions such as SSNHL.

## Limitation

In defining SVD, our approach relied on established parameters from previous studies, allowing for comprehensive comparisons within the SSNHL population. However, it is essential to acknowledge certain limitations pertaining to the analysis of specific vascular diseases, including ICH, CADASIL, and Sickle Cell Disorder. The dataset's insufficient representation of these conditions restricted their detailed examination within the context of SVD and SSNHL. Moreover, the utilization of the HIRA database, while advantageous in its coverage of 98% of the South Korean population, is not without drawbacks. Notably, the absence of detailed test results poses a challenge in refining criteria, particularly in cases like pure tone audiometry tests. While the presence or absence of a test is recorded, the lack of result documentation limits our ability to establish criteria based on specific hearing thresholds at different frequencies. Consequently, this impediment precluded the application of a precise clinical definition. These methodological considerations underscore the need for cautious interpretation and highlight the inherent limitations in our study design. While the HIRA database provides a valuable resource for population-level analyses, future research efforts should address these constraints to enhance the comprehensiveness and accuracy of investigations into the relationship between vascular diseases and SSNHL.

## Conclusion

We identified SVD as a possible cause of SSNHL. Previous studies have defined SVD as a condition that includes various vascular diseases. Using the ratio of each vascular disease, SVD was redefined as any vascular disease with a greater influence on the SSNHL group. The SVD group created in this manner showed various influences within the SSNHL group, and it was confirmed that the duration of SSNHL increased only in the presence of SVD. In particular, HR was higher in the group that developed SVD after the onset of SSNHL. SVD also affected the recurrence of SSNHL, and the recurrence rate was 1.312 times higher in the group with SVD. An analysis confirming the time to recurrence showed that recurrence occurred more rapidly in the SVD group. The findings highlight the intricate relationship between SVD and SSNHL, emphasizing the potential impact of vascular factors on SSNHL duration and recurrence. Further research, especially considering the limitations, is warranted to unravel the underlying mechanisms and explore clinical interventions for SSNHL patients with associated vascular conditions.

## Supporting information

**S1 Table. Classification codes of anti-hypertensive drugs.**
(PDF)

**S2 Table. Classification codes of diabetes mellitus drugs.**
(PDF)

**S3 Table. Classification codes of dyslipidemia drugs.**
(PDF)

**S4 Table. Procedure code in small-vessel disease.**
(PDF)

**S5 Table. Drug code & procedure code in sudden sensorineural hearing loss.**
(PDF)

## Author Contributions

**Conceptualization:** Tae Hoon Kong, Young Joon Seo.

**Data curation:** Chul Young Yoon, Junhun Lee.

**Formal analysis:** Chul Young Yoon, Junhun Lee, Tae Hoon Kong.

**Funding acquisition:** Young Joon Seo.

**Investigation:** Chul Young Yoon.

**Methodology:** Chul Young Yoon, Tae Hoon Kong.

**Project administration:** Tae Hoon Kong, Young Joon Seo.

**Resources:** Chul Young Yoon, Junhun Lee.

**Software:** Chul Young Yoon, Junhun Lee.

**Supervision:** Young Joon Seo.

**Validation:** Chul Young Yoon, Junhun Lee.

**Visualization:** Chul Young Yoon, Junhun Lee.

**Writing – original draft:** Chul Young Yoon.

**Writing – review & editing:** Chul Young Yoon, Tae Hoon Kong, Young Joon Seo.

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
