## [Decision Letter · Decision Letter 0]

12 Jan 2024

PONE-D-23-35617Importance of Small Vessel Disease as a possible cause of Sudden Sensorineural Hearing LossPLOS ONE

Dear Dr. Seo,

Thank you for submitting your manuscript to PLOS ONE. After careful consideration, we feel that it has merit but does not fully meet PLOS ONE’s publication criteria as it currently stands. Therefore, we invite you to submit a revised version of the manuscript that addresses the points raised during the review process.

We look forward to receiving your revised manuscript.

Kind regards,

Abdullah M. Mutawa, Ph.D

Academic Editor

PLOS ONE

Journal Requirements:

3. In the online submission form, you indicated that "Data cannot be shared publicly because of National health insurance data from Korea. Data are available from the National health insurance service Institutional Data Access / Ethics Committee (contact via https://nhiss.nhis.or.kr/bd/ay/bdaya001iv.do) for researchers who meet the criteria for access to confidential data."

4. We note you have included a table to which you do not refer in the text of your manuscript. Please ensure that you refer to Table 2 in your text; if accepted, production will need this reference to link the reader to the Table.

Reviewers' comments:

Reviewer's Responses to Questions

**Comments to the Author**

1. Is the manuscript technically sound, and do the data support the conclusions?

Reviewer #1: Partly

Reviewer #2: Yes

2. Has the statistical analysis been performed appropriately and rigorously? 

Reviewer #1: No

Reviewer #2: No

3. Have the authors made all data underlying the findings in their manuscript fully available?

Reviewer #1: No

Reviewer #2: Yes

4. Is the manuscript presented in an intelligible fashion and written in standard English?

Reviewer #1: No

Reviewer #2: Yes

5. Review Comments to the Author

Reviewer #1: In their paper “Importance of Small Vessel Disease as a possible cause of Sudden Sensorineural

Hearing Loss”, the authors have attempted to describe the frequency and possible causes of postoperative bleeding after endoscopic nasal and sinus surgery. Though the attempt is laudable, I have several issues with the paper.

1- In the introduction, authors noted the spontaneous recovery rate 32–65% for SSNHL. In Clinical Practice Guideline: Sudden Hearing Loss (2019), it is noticed that Clinical experience shows that these numbers may be an overestimation.

2- Following paragraph, especially ref 13 is unnecessary:

In 2018, a study investigating the relationship between ischemic stroke and SSNHL was conducted8. In 2020, a study was conducted to investigate the relationship between plasma serotonin elevation, metabolic syndrome, and SSNHL9, 10. In 2021, a study was conducted to investigate the relationship between stroke, atherosclerosis, and SSNHL11. In 2022, a study investigating the relationship between cardiovascular risk factors and SSNHL was conducted12. South Korea's Health Insurance Review and Assessment Service (HIRA) collects claims data generated when citizens visit medical institutions in accordance with their health insurance system13. The HIRA database is available to researchers on a limited basis. Korean researchers are continuously conducting research on vascular diseases using the HIRA database4, 14-19. However, no study has confirmed the extent to which vascular diseases cause SSNHL.

3- Self-citations of authors could be observed in ref 13, 28, 29, 30, and 31.

4- This sentence is unclear: “This study was conducted with data access permission from HIRA from 2022-08-04 to 2022-12-31.”

5- What is answer of the first aim? According to Figure 1, 123832 out of 319569 patients with SSNHL (38.7%) have SVD. Which proportion of general population have SVD.

6- The second aim isn’t logical.

“Subject 2. To determine the effect of SVD on illness duration in patients with SSNHL” Therefore, In Table 4 and Figure 2, results are not realistic. We know at least one thirds of patients don’t recover. But we observe that approximate zero percent of patients have SSNHL after 100 days in Figure 2. Also, it is expected HR in patients with SVD before onset of SSNHL is more than patients with SVD after onset of SSNHL. In table 2, the HRs are depicted 1.045 and 1.234, respectively.

7- The definition of duration of treatment for SSNHL is not correct.

8- Table 2 is vague. Table 3 is unnecessary.

9- Why total number of patients in table 2 and 3 is less than 319569.

10- Table 5, data in unclear. How many patients with or without SVD had recurrence? We could consider number of patients with recurrence in patients with and without SVD to be 4093 and 15848, respectively; and number of patients without recurrence in patients with and without SVD to be 34942 and 104826, respectively. Then OR would be 1.34 (95%CI 1.29-1.39).

11- What was the mean and median time to recurrence? In figure 2-B, we expect upward sloping of Kaplan-Meier curve with time.

12- Discussion is weak. The first paragraph is unnecessary and references 28-31 are irrelevant.

13- Omit the duplicated sentences in the manuscript.

14- Interpret results in the discussion part.

15- Specify limitation of this study.

Reviewer #2: This is an interesting paper on the impact of SVD on SSNHL using national scale data. There have been many studies suggesting that vascular problems play a significant role in the onset of SSNHL, a point with which I agree. However, I would like to make a few suggestions and ask some questions about this study.

<introduction>

"Previous studies have suggested possible causes of SSNHL, including vascular disorders, viral infections, and bacterial infections(1,5-7)"

In the introduction, it would be beneficial to suggest a broader range of causes of SSNHL. Autoimmune diseases could be a prime example.

<method>

"This study was conducted with data access permission from HIRA from 2022-08-04 to 2022-12-31"

As far as I know, HIRA requires you to specify the research approval number in the study paper. Please include research number of HIRA here.

Duration -

Could the duration vary more depending on the unique treatment style of each hospital? Given that this is a nationwide data, could the tendencies of physicians or the routine treatment policies of institutions act as a bias?

"the difference between each diagnosis date was less than 365 days. Recurrence of SSNHL was defined as a difference of > 365 days from the date of diagnosis,"

How do you filter out someone who has a follow-up at the same institution after a year with 91.2 code? Did you only use cases that satisfied the same operational definition for the recurring SSNHL(91.2)?

"The statistical program used for the analysis was SAS version 9.4 (SAS Institute Inc., Cary, NC, USA)."

Please create a separate subsection and write more about the statistical processing techniques.

What is Propensity Score Matching and why was it implemented? It should be specified in the statistical analysis methodology. It needs to be mentioned for those who are not familiar with statistical methodologies, including the reviewer.

Table 1

Please spell out the abbreviations in the table description.

Operational definition section

What is the reason for using only the diagnosis code in certain diseases and conditions other than drugs code or other features ?

<results>

How about listing the demographics of actual SSNHL patients to Table 2 or a new table?

Were there no demographic features to write other than the male-female ratio?

Table 3

"SVD ratio in National record"

Is this data newly extracted for each diagnosis name between 2010 and 2020? When was the targeted national population? And this content did not seem to be specified in the method section

"Confidence Interval of SVD ratio in SSNHL"

Is it correct to only write CI? Shouldn't you also write the Ratio value?

Did you calculate SVD regardless of before and after the onset of SSNHL here?

Table 5

Isn't the recurrence higher than the generally known recurrence rate of sudden sensorineural hearing loss?

"Ko HY, Nam HJ, Kim MH. A Nationwide Population-Based Study for the Recurrence and Comorbidities in Sudden Sensorineural Hearing Loss. Laryngoscope. 2023 Sep 22."

In "time to recurrence", shouldn't the sum of N be 34,942? "(4093+15848 =????)"</results></method></introduction>

6. PLOS authors have the option to publish the peer review history of their article (what does this mean?). If published, this will include your full peer review and any attached files.

Reviewer #1: No

Reviewer #2: No

---

## [Author Response · Author response to Decision Letter 0]

30 Jan 2024

Response to Reviewer 1 Comments

We have restructured the 'MATERIALS and METHODS' and 'RESULTS' sections based on common feedback from reviewers. In the 'MATERIALS and METHODS' section, the experimental procedure, utilized statistics, flowchart, and Table 2 have been revised to provide a clearer presentation of how the experiment was conducted. The 'RESULTS' section has been labeled according to subjects.

1- In the introduction, authors noted the spontaneous recovery rate 32–65% for SSNHL. In Clinical Practice Guideline: Sudden Hearing Loss (2019), it is noticed that Clinical experience shows that these numbers may be an overestimation.

We added the point that there is a report on overestimation.

2- Following paragraph, especially ref 13 is unnecessary:

In 2018, a study investigating the relationship between ischemic stroke and SSNHL was conducted8. In 2020, a study was conducted to investigate the relationship between plasma serotonin elevation, metabolic syndrome, and SSNHL9, 10. In 2021, a study was conducted to investigate the relationship between stroke, atherosclerosis, and SSNHL11. In 2022, a study investigating the relationship between cardiovascular risk factors and SSNHL was conducted12. South Korea's Health Insurance Review and Assessment Service (HIRA) collects claims data generated when citizens visit medical institutions in accordance with their health insurance system13. The HIRA database is available to researchers on a limited basis. Korean researchers are continuously conducting research on vascular diseases using the HIRA database4, 14-19. However, no study has confirmed the extent to which vascular diseases cause SSNHL.

We deleted.

3- Self-citations of authors could be observed in ref 13, 28, 29, 30, and 31.

Ref 10 refers to our previous study, essential for defining SSNHL within big data. Therefore, self-citation was unavoidable. The remaining references have been removed.

4- This sentence is unclear: “This study was conducted with data access permission from HIRA from 2022-08-04 to 2022-12-31.”

We added this sentence: “We obtained data access permission from HIRA, and the period during which we could use the data was from 2022-08-04 to 2022-12-31. The approval numbers for this study obtained from HIRA are M20210430249 and M20221005002.”

5- What is answer of the first aim? According to Figure 1, 123832 out of 319569 patients with SSNHL (38.7%) have SVD. Which proportion of general population have SVD.

The results for Subject 1 (first aim) can be found in Table 3. The paragraph above contains the following information: “It is possible to verify each SVD individually in the NHI database, but it is not possible to identify duplicate patients. Therefore, the expression is presented as a ratio for each SVD condition.”

6- The second aim isn’t logical.

“Subject 2. To determine the effect of SVD on illness duration in patients with SSNHL”

Therefore, In Table 4 and Figure 2, results are not realistic. We know at least one thirds of patients don’t recover. But we observe that approximate zero percent of patients have SSNHL after 100 days in Figure 2. Also, it is expected HR in patients with SVD before onset of SSNHL is more than patients with SVD after onset of SSNHL. In table 2, the HRs are depicted 1.045 and 1.234, respectively.

Our definition of the treatment period for our patients is from the first diagnosis date to the last diagnosis date. Therefore, the results in Figure 2A signify that the respective patient is no longer under observation. In our data, this implies that the patient did not seek further hospital visits with the H91.2 code within one year. Typically, within 3 months, a patient's treatment concludes, transitioning into a phase of management. Furthermore, as we also address the keyword 'recurrence,' instances where the gap between H91.2 codes is more than one year are separately classified as recurrences, and subsequent outcomes are excluded from Subject 2.

7- The definition of duration of treatment for SSNHL is not correct.

Retrospectively collected data, especially health insurance data, has the advantage of being able to consider a vast amount of information. However, a drawback is that it often lacks the opportunity to utilize the clinical definitions commonly used in the field. If the 'duration of treatment for SSNHL' is meant to represent what we defined as 'the first diagnosis date to the last diagnosis date within 365 days,' then our definition attempts to align with the available resources in the data as consistently as possible.

8- Table 2 is vague. Table 3 is unnecessary.

We have revised Table 2, and Table 3 presents the results for Subject 1.

9- Why total number of patients in table 2 and 3 is less than 319569.

Since PSM was conducted for each subject separately, the number of patients varies by subject, as described in the 'MATERIALS and METHODS' section.

10- Table 5, data in unclear. How many patients with or without SVD had recurrence? We could consider number of patients with recurrence in patients with and without SVD to be 4093 and 15848, respectively; and number of patients without recurrence in patients with and without SVD to be 34942 and 104826, respectively. Then OR would be 1.34 (95%CI 1.29-1.39).

Due to the matching results, the number of patients varies for each subject. In Subject 3, there are 34,942 patients with recurrence regardless of SVD, and 104,826 patients without recurrence. We utilized third-party data, and we no longer have access to it. Therefore, we do not know the number of patients with SVD regardless of recurrence in that particular dataset. The original number of patients before PSM is as follows, as shown in the table below.

SVD Yes No

Recurrence Yes 15,848 19,094

 No 110,963 173,664

11- What was the mean and median time to recurrence? In figure 2-B, we expect upward sloping of Kaplan-Meier curve with time.

In SVD, Mean time: 37.407month, Median time: 30 month. In No SVD, Mean time: 35.836month, Median time: 29 month. We have incorporated it into Table 5-B.

12- Discussion is weak. The first paragraph is unnecessary and references 28-31 are irrelevant.

We deleted and strengthening the discussion.

13- Omit the duplicated sentences in the manuscript.

14- Interpret results in the discussion part.

15- Specify limitation of this study.

We checked it.

 

Response to Reviewer 2 Comments

We have restructured the 'MATERIALS and METHODS' and 'RESULTS' sections based on common feedback from reviewers. In the 'MATERIALS and METHODS' section, the experimental procedure, utilized statistics, flowchart, and Table 2 have been revised to provide a clearer presentation of how the experiment was conducted. The 'RESULTS' section has been labeled according to subjects.

"Previous studies have suggested possible causes of SSNHL, including vascular disorders, viral infections, and bacterial infections (1,5-7)"

In the introduction, it would be beneficial to suggest a broader range of causes of SSNHL. Autoimmune diseases could be a prime example.

We added this sentence: “Previous studies have suggested possible causes of SSNHL, including vascular, ear & endocrinal disorder, infection, autoimmune, cancer, trauma, and drugs adverse effect”

We added this reference: Tripathi P, Deshmukh P. Sudden Sensorineural Hearing Loss: A Review. Cureus. 2022 Sep 22;14(9):e29458. doi: 10.7759/cureus.29458. PMID: 36299969; PMCID: PMC9587755.

"This study was conducted with data access permission from HIRA from 2022-08-04 to 2022-12-31"

As far as I know, HIRA requires you to specify the research approval number in the study paper. Please include research number of HIRA here.

We added this sentence: “We obtained data access permission from HIRA, and the period during which we could use the data was from 2022-08-04 to 2022-12-31. The approval numbers for this study obtained from HIRA are M20210430249 and M20221005002.”

Duration -

Could the duration vary more depending on the unique treatment style of each hospital? Given that this is a nationwide data, could the tendencies of physicians or the routine treatment policies of institutions act as a bias?

We agree with that perspective. However, given the nationwide data, the unique treatment styles of each hospital and the preferences of individual doctors are expected to regress to the national average.

"the difference between each diagnosis date was less than 365 days. Recurrence of SSNHL was defined as a difference of > 365 days from the date of diagnosis,"

How do you filter out someone who has a follow-up at the same institution after a year with 91.2 code? Did you only use cases that satisfied the same operational definition for the recurring SSNHL (91.2)?

The recurrence was defined as cases where the time between the last treatment and the subsequent diagnosis (previous H91.2 code and the next H91.2 code) exceeded 365 days, even if it was within the same institution.

When defining recurrence, we did not consider a recurrence if the next follow-up of SSNHL care occurred after 1 year. We will summarize these limitations in the discussion regarding recurrence: “While efforts were made to define the duration of SSNHL and recurrence as objectively as possible, bias inevitably arises due to the researcher's influence. Recurrence was defined as cases where the time between diagnoses was within 365 days; however, in some instances, patients may seek hospital care for the management of SSNHL beyond the 365 days after the initial diagnosis, indicating a conceptual challenge in defining recurrence.”

Recurrent SSNHL adhered to the same definition across all instances.

"The statistical program used for the analysis was SAS version 9.4 (SAS Institute Inc., Cary, NC, USA)."

Please create a separate subsection and write more about the statistical processing techniques.

What is Propensity Score Matching and why was it implemented? It should be specified in the statistical analysis methodology. It needs to be mentioned for those who are not familiar with statistical methodologies, including the reviewer.

Due to common feedback from reviewers, I have restructured the organization of the methods and results sections in the main text. To enhance clarity in study design, I moved the results of Propensity Score Matching (PSM) to the methods section. Detailed explanations and outcomes of PSM can be found in the "Experimental Design and Statistics" section.

Table 1

Please spell out the abbreviations in the table description.

We checked it.

Operational definition section

What is the reason for using only the diagnosis code in certain diseases and conditions other than drugs code or other features?

I'm not sure if I understood the question correctly. When defining all diseases in our study, we took into account disease codes, prescription histories, and medical test records. The reason for not considering test results or other conditions in the clinical definition is that the health insurance data did not include such information. This limitation stems from the absence of these test results or conditions in the health insurance data, and it represents a constraint in conducting retrospective studies utilizing big data.

How about listing the demographics of actual SSNHL patients to Table 2 or a new table?

Were there no demographic features to write other than the male-female ratio?

When examining the results of Propensity Score Matching (PSM), we opted not to include demographic statistics as the most crucial aspect was demonstrating a lack of significant differences in proportions between the two groups. Aside from gender and age, no additional significant features were identified. 

Table 3

"SVD ratio in National record"

Is this data newly extracted for each diagnosis name between 2010 and 2020? When was the targeted national population? And this content did not seem to be specified in the method section

We have revised the 'Methods' and 'Results' sections. The target population consists of the entire population from 2010 to 2020. Figure 1 has been added to the revised 'Methods' section. Here is an excerpt: 'In Subject 1, we compare the ratio of SVD between the entire population of South Korea and the SSNHL patient group from 2010 to 2020.'

"Confidence Interval of SVD ratio in SSNHL"

Is it correct to only write CI? Shouldn't you also write the Ratio value?

Did you calculate SVD regardless of before and after the onset of SSNHL here?

We have included the ratio and calculated SVD regardless of pre- and post-correlation.

Table 5

Isn't the recurrence higher than the generally known recurrence rate of sudden sensorineural hearing loss?

"Ko HY, Nam HJ, Kim MH. A Nationwide Population-Based Study for the Recurrence and Comorbidities in Sudden Sensorineural Hearing Loss. Laryngoscope. 2023 Sep 22."

The definition of recurrence in retrospective data varies depending on the perspective of viewing episodes. In our study, we conservatively defined recurrence, considering an ongoing episode if the distance between diagnoses was within one year. Therefore, we deemed a recurrence to occur only if the H91.2 code was not observed for more than one year. Additionally, the method of defining SSNHL may differ. While considering diagnosis codes, pure-tone audiometry tests, and steroid prescriptions, the inclusion criteria for pure-tone audiometry codes or the extent of including steroid codes can impact the patient count. Even the inclusion of years in the analysis can lead to variations in results.

In "time to recurrence", shouldn't the sum of N be 34,942? "(4093+15848 =????)"

Since PSM was conducted for each subject separately, the number of patients varies by subject, as described in the 'MATERIALS and METHODS' section.

---

## [Decision Letter · Decision Letter 1]

4 Apr 2024

Importance of Small vessel disease as a possible cause of Sudden sensorineural hearing loss

PONE-D-23-35617R1

Dear Dr. Seo,

We’re pleased to inform you that your manuscript has been judged scientifically suitable for publication and will be formally accepted for publication once it meets all outstanding technical requirements.

Kind regards,

Manoj Diwakar

Academic Editor

PLOS ONE

Additional Editor Comments (optional):

Although Reviewer 1 recommended amendment of the tables and figures, I feel the manuscript is suitable to proceed without further amendment.

Reviewers' comments:

Reviewer's Responses to Questions

**Comments to the Author**

1. If the authors have adequately addressed your comments raised in a previous round of review and you feel that this manuscript is now acceptable for publication, you may indicate that here to bypass the “Comments to the Author” section, enter your conflict of interest statement in the “Confidential to Editor” section, and submit your "Accept" recommendation.

Reviewer #1: (No Response)

Reviewer #2: All comments have been addressed

2. Is the manuscript technically sound, and do the data support the conclusions?

Reviewer #1: Partly

Reviewer #2: Yes

3. Has the statistical analysis been performed appropriately and rigorously? 

Reviewer #1: I Don't Know

Reviewer #2: Yes

4. Have the authors made all data underlying the findings in their manuscript fully available?

Reviewer #1: No

Reviewer #2: Yes

5. Is the manuscript presented in an intelligible fashion and written in standard English?

Reviewer #1: Yes

Reviewer #2: Yes

6. Review Comments to the Author

Reviewer #1: (No Response)

Reviewer #2: As the reviewer, I have thoroughly examined the revised version of your manuscript. I am pleased to see that you have carefully addressed and incorporated all the points I mentioned in my previous review. The modifications you have made have significantly improved the clarity, coherence, and overall quality of the paper.

7. PLOS authors have the option to publish the peer review history of their article (what does this mean?). If published, this will include your full peer review and any attached files.

Reviewer #1: No

Reviewer #2: No
